# The Moderating Effect of Self-Efficacy on Physical Function, Aging Anxiety, and Active Aging in Community-Dwelling Older Adults

**DOI:** 10.3390/healthcare13020108

**Published:** 2025-01-08

**Authors:** Myunghwa Oh, Gyeong-A Park

**Affiliations:** 1Department of Occupational Therapy, Dongshin University, Naju 58245, Republic of Korea; mhoh@dsu.ac.kr; 2Department of Occupational Therapy, Chosun University, Gwangju 61452, Republic of Korea

**Keywords:** self-efficacy, physical function, aging anxiety, active aging, older adults

## Abstract

**Objectives:** Aging older adults experience psychological anxiety along with declines in physical function, which decreases the level of active aging and physical activity participation. The purpose of this study was to identify the moderating effect of self-efficacy on active aging to improve the occupational participation of community-dwelling older adults. **Methods:** A cross-sectional survey was conducted among older adults in Gwangju and Jeollanamdo using the snowball sampling method. The final sample consisted of 280 adults. The survey included demographic data, the Outpatient Physical Therapy Improvement in Movement Assessment Log, the Anxiety about Aging Scale, the General Self-Efficacy Scale, and the Active Aging Scale, respectively. Using the Hayes PROCESS macro (Models 3.3), moderated effect analyses were performed. **Results:** A total of 307 participants (aged 65–92 years) and 280 data points were used in the final analysis after excluding 27 incomplete data points. The physical function of older adults has a positive effect on active aging, aging anxiety significantly predicts negative effects, and self-efficacy shows a moderating effect on the relationship between physical function, aging anxiety, and active aging. **Conclusions:** This study shows the moderating effect of self-efficacy on the relationship between physical function, aging anxiety, and active aging in older people. These results suggest that a psychological support program to promote self-efficacy is an important resource as a community support system to prevent decline in occupational participation due to physical function decline and aging anxiety and to improve active aging in older people.

## 1. Introduction

In 2022, the older adult population in the world accounted for approximately 10%, and the older adult population is increasing mainly in countries with high levels of economic development [1]. Korea will have 19.2% older adults in 2024 and is expected to reach a super-aged society by 2025 [2]. Therefore, personal responses and social change for active and safe aging are challenges facing community rehabilitation areas.

The World Health Organization defines the concept of ‘active aging’ as ‘the process of optimizing opportunities for health, participation, and security in order to enhance quality of life as people age’ [3]. Expectations of active aging are a priority when preparing for old age. The general concept of active aging is that the degree of disability is low, the level of physical fitness is high, and participation in life is maintained with high cognitive function, positive mood, and stress-coping ability [4]. Physical function and psychological factors are considered important in preparing for active aging in old age. Therefore, to increase life satisfaction in old age, individual efforts are required to maintain health, and the need for psychological and social support is increasing so that older adults can seek opportunities for more independent and active activities.

A feature of the aging process can be seen as a decrease in physical function, such as a decrease in physical ability and strength [5]. In the aging process, the physical function of older adults is a direct factor that affects their adaptation to the environment and their performance of activities and determines whether they can act independently by considering the qualitative aspects of their old age [6]. Generations entering old age attempt to find their roles through active activities and new challenges [7], and an active lifestyle can mitigate the aging process [8]; physically active older adults have better health and vitality [9]. This indicates that physical function affects active aging by acting as a behavioral pattern and psychological factor to the maintenance of health among older adults.

In relation to aging, depression, anxiety, and dementia are likely to cause psychological problems, which can deteriorate the quality of life of older adults [10]. Aging anxiety is a negative emotion associated with aging, including the fear of aging and feelings of physical, mental, and personal loss [11]. It is a complex concept that includes psychological and social loss as well as physical health and economic anxiety [12]. This indicates that aging anxiety affects activities as an important factor that mediates attitudes and behaviors toward older adults or adaptation to aging [13]. Therefore, aging anxiety affects the quality of life and happiness of older adults and affects overall activities.

Self-efficacy is an important internal resource in terms of adapting to challenges and changes in old age and the ability to continue growing later in life [14]. Older adults decline in function as they age, but not all psychosocial functions are impaired. A developmental process that can lead to positive changes in one’s ability can also help develop a new ability to compensate for a loss in another area [15]. We can improve our self-efficacy by imagining that we can easily overcome fearful behaviors or difficulties, encourage disappointed people to believe that they are more capable than they think, and encourage psychological interventions that encourage them to achieve self-enhancing success [16]. Self-efficacy is the most influential factor in changing patient health behaviors and also acts as a strategy to promote motivation for rehabilitation [17]. Therefore, it is necessary to improve self-efficacy as a measure to prepare for a high-quality life in old age by encouraging older adults to participate in activities to maintain health and live with an active attitude toward safety. In old age, it is necessary to self-regulate self-efficacy and confirm one’s beliefs. Through this, it is possible to prepare an appropriate compensation strategy and perform an effective aging design while limiting the scope of activities.

Looking at previous studies, in the relationship between physical activity level and active aging of older adults, when physical activity level and physical strength were high, the level of active aging was also high [18,19], and the quality of life of older adults with high self-efficacy appears to be high [20]. Aging anxiety negatively affects self-efficacy, self-fulfillment, and successful aging. It has been reported that factors affecting successful aging include physical, mental, and social health [14,21]. Self-efficacy affects the control of human psychology and function. In a study on the factors that affect the satisfaction of life of older adults, self-efficacy was found to be an important variable that had a positive effect [14], and it was reported that it was also used as an intervention strategy to improve functional status in a study on older adults [22].

The previous literature has shown that physical function and aging anxiety in older adults are factors that affect active aging. In addition, internal psychological resources such as self-efficacy can improve older adults’ participation in activities and support healthy and safe old age. However, there is a lack of research on the intervention of self-efficacy on the relationship between physical function, aging anxiety, and active aging of older adults. Therefore, this study aimed to identify the moderating effect of self-efficacy that promotes active aging for older adults with physical difficulties and aging anxiety. Through this, we would like to suggest the necessity of developing a psychological support program for active aging of older adults.

## 2. Materials and Methods

### 2.1. Type of Study

This study was a cross-sectional study designed to examine the relationships between physical function, aging anxiety, active aging, and self-efficacy. The type of sampling used to collect data was snowball sampling. The subjects were older adults aged 65 or older living in Gwangju and Jeollanamdo, and the data were collected through questionnaires, from August 2022 to March 2023. This study was approved by the Institutional Review Board of Dongshin University (IRB No. 201905-SB-017).

### 2.2. Study Participants

Participants were recruited via flyers. The number of sample groups was calculated using the G* power 3.1 program, with an effect size of 0.15, significance probability of 0.05, power of 95%, and 3 independent variables; the minimum required sample size determined was 129 [23]. The total number of participants collected through the published flyers was 307; however, 27 questionnaires including insincere responses and missing values were excluded, and 280 were used in the final analysis [24]. The survey was conducted on subjects who understood the research plan and filled out a written consent form. Older adults with cognitive impairment or difficulty in physical function were excluded from the criteria.

### 2.3. Participant Recruitment Procedure

Participants in this study were recruited primarily by sending out informational materials containing the purpose, subjects, content, procedures, and consent forms to senior health centers, senior welfare centers, and public health centers in Gwangju and Jeollanamdo, South Korea. The researcher visited the institution to obtain written consent from the participants, explained the study, conducted a survey with the participants, who agreed to participate in the study based on their personal freedom and autonomy, and obtained their signatures.

### 2.4. Measurement Instruments

Physical functions: Physical function in older adults was assessed using the Outpatient Physical Therapy Improvement in Movement Assessment Log (OPTIMAL). The OPTIMAL is a self-reported questionnaire that measures difficulty and self-confidence in performing 21 movements that are necessary for accomplishing activities of daily living. Each item was rated on a 5-point ordinal scale ranging from 1 (able to without any difficulty) to 5 (unable to do). The total scores can range from 21 to 105, and a higher score indicates greater impairment in physical function [25]. In this study, all items were reversed and analyzed in the statistical analysis; the higher the score, the better the physical function was interpreted. Cronbach’s *α* in Guccione and colleagues was 0.85~0.95 and Cronbach’s *α* in this study was 0.96.

Aging anxiety: Aging anxiety was assessed using the Korean version of the Anxiety about Aging Scale (AAS). The AAS developed by Lasher and Faulkender and translated into Korean by Kim was used. It is a 20-item self-report scale with 13 positive and 7 negative items. Each element was rated on a 5-point Likert scale with the degree of aging anxiety comprising 1 point ‘not at all’ to 5 points ‘definitely yes’ [12,26]. In this study, the negative items were reversed and reflected in the statistical analysis; lower scores indicated a lower level of aging anxiety. Cronbach’s *α* for Kim was 0.86. Cronbach’s *α* in this study was 0.88.

Self-Efficacy: Self-efficacy was assessed using the Korean version of the General Self-Efficacy Scale (GSES). The GSES was developed by Sherer et al. and translated into Korean by Oh. It is a 17-item self-report scale designed to assess the degree of self-efficacy and consists of 6 positive and 11 negative [27,28]. Each item was rated on a 5-point Likert scale, with the degree of self-efficacy ranging from 1 point ‘not at all true’ to 5 points ‘exactly true’. In this study, negative items were reversed and reflected in the statistical analysis, and higher scores indicated higher self-efficacy. Cronbach’s *α* of Oh was 0.80. Cronbach’s *α* in this study was 0.95.

Active Aging: Active aging was assessed using the Active Aging Scale. The Active Aging Scale developed by Ryu et al. tests the validity of older adults to measure the degree of active aging among older adults [29]. It consists of 51 items, including 30 items of participation, 8 items of safety, and 13 items of health. Each item was rated on a self-reported 5-point Likert scale, with the degree of active aging being composed of 1 point ‘strongly disagree’ to 5 points ‘strongly agree’, and a higher score indicated a higher level of active aging. Cronbach’s *α* of Ryu and colleagues was 0.62~0.89. Cronbach’s *α* of this study was 0.93.

### 2.5. Statistical Analysis

The data collected in this study were analyzed using SPSS Statistics 27.0 and Process Macro (3.3 version) with statistical significance at *p* ≤ 0.05; frequency and descriptive statistics and percentages were analyzed for the demographic variables. Means, standard deviations, reliability values (that is Cronbach alphas), and normality checks were calculated for all main study variables. The correlation between the major variables was analyzed using Pearson’s correlation coefficients, and the PROCESS macro was used to specifically examine the moderating and conditional effects of self-efficacy on the relationship between physical function, aging anxiety, and active aging [30]. Process macromodel 1 was applied to analyze the moderating effect, and mean centering was performed to reduce multicollinearity between variables. A pick-a-point approach was used to test the inference of the moderating effect, and the statistical significance of the moderating effect was tested by bootstrapping.

## 3. Results

### 3.1. Demographic Data

Our study included 280 older people living in Gwangju and Jeollanamdo, South Korea. The age ranged from 65 to 92 years (M = 74.84, SD = 7.0). The participants consisted of 151 (53.9%) males and 129 (46.1%) females. Among the elderly, 12 (4.3%) were uneducated and 92 (32.9%) were high school graduates with academic education. The detailed characteristics of the participants are presented in Table 1.

### 3.2. Descriptive Statistics and Correlations of Measured Variables

The descriptive statistics and correlations of the measured variables are presented in Table 2. The mean scores for physical function, aging anxiety, self-efficacy, and active aging were 66.20 (*SD* = 16.86, range = 30.00–105.00), 58.82 (*SD* = 10.02, range = 22–80), 59.22 (*SD* = 10.81, range = 33–85), and 157.68 (*SD* = 24.70, range = 80–234), respectively. For the correlations, physical function was positively correlated with self-efficacy and active aging (*r* = 0.172, *r* = 0.611, *p* < 0.001) and inversely correlated with aging anxiety (*r* = −0.574, *p* < 0.001). Furthermore, aging anxiety was negatively correlated with self-efficacy and active aging (*r* = −0.160, *r* = −0.595, *p* < 0.01), whereas self-efficacy was positively correlated with active aging (*r* = 0.360, *p* < 0.01).

### 3.3. Moderating Effects of Self-Efficacy on Physical Function and Active Aging

To examine whether self-efficacy moderates the relationship between physical function and active aging, the PROCESS micromodel was used to analyze the moderating effect (Table 3). The effect of physical function on active aging was significant (*B* = 0.725, *p* < 0.001), and the effect of self-efficacy on active aging was significant (*B* = 0.804, *p* < 0.001). In other words, the higher the physical function and self-efficacy, the higher the level of active aging. Physical function and self-efficacy were mean-centered and analyzed simultaneously. As a result, the moderating effect of the interaction term was significant (*B* = 0.029, *p* < 0.001). In addition, *LLCI* and *ULCI* (confidence interval 95%) did not include ‘0’, showing significant results, and self-efficacy showed a moderating effect in the relationship between physical function and active aging. In other words, physical function positively influenced active aging, and self-efficacy positively influenced this effect. Therefore, the higher the physical function level, the higher the level of active aging, and self-efficacy increases this effect (Figure 1).

### 3.4. Conditional Effects of Self-Efficacy on Physical Function and Active Aging

The conditional effect of self-efficacy on the relationship between physical function and active aging was examined (Table 4). The self-efficacy value was divided into low (−1SD), mean, and high (−1SD) groups, and there was no ‘0’ between the lower and upper limits of all three groups, so it was statistically significant. In other words, the lower the level of self-efficacy (*B* = 0.376, *p* < 0.001), the higher the physical function, and the higher the level of active aging. When the level of self-efficacy was mean (*B* = 0.747, *p* < 0.001), the higher the physical function, the higher the level of active aging. The higher the level of self-efficacy (*B* = 1.034, *p* < 0.001), the higher the physical function, and the higher the level of active aging. This showed that as the level of self-efficacy increased, the influence of physical function on active aging increased.

### 3.5. Moderating Effects of Self-Efficacy on Aging Anxiety and Active Aging

To examine whether self-efficacy moderates the relationship between aging anxiety and active aging, the PROCESS macromodel was used to analyze the moderating effect (Table 5). The effect of aging anxiety on active aging was significant (*B* = −1.112, *p* < 0.001), and the effect of self-efficacy on active aging was significant (*B* = 0.803, *p* < 0.001). In other words, the higher the aging anxiety level, the lower the level of active aging and the opposite effect of self-efficacy. Aging anxiety and self-efficacy were mean-centered and analyzed simultaneously. As a result, the moderating effect of the interaction term was significant (*B* = −0.048, *p* < 0.001). In addition, *LLCI* and *ULCI* (confidence interval 95%) did not include ‘0’, showing significant results, and self-efficacy showed a moderating effect on the relationship between aging anxiety and active aging. In other words, aging anxiety has a negative effect on active aging, whereas self-efficacy has a positive effect. As a result, the level of active aging decreased as aging anxiety increased, but active aging increased due to the moderating effect of self-efficacy (Figure 2).

### 3.6. Conditional Effects of Self-Efficacy on Aging Anxiety and Active Aging

The conditional effect of self-efficacy on the relationship between aging anxiety and active aging was examined (Table 6). The self-efficacy value was divided into low (−1SD), mean, and high (+1SD) groups, and there was no ‘0’ between the lower and upper limits of all three groups, so it was statistically significant. In other words, the lower the level of self-efficacy (*B* = −0.520, *p* < 0.001), the lower the aging anxiety, and the higher the level of active aging. When the level of self-efficacy was mean (*B* = −1.150, *p* < 0.001), the lower the aging anxiety, and the higher the level of active aging. The higher the level of self-efficacy (*B* = −1.636, *p* < 0.001), the lower the aging anxiety, and the higher the level of active aging. This showed that as the level of self-efficacy increased, the influence of aging anxiety on active aging decreased.

## 4. Discussion

The purpose of the present study was to examine the effect of physical function and aging anxiety on active aging among older adults and the moderating effect of self-efficacy on physical function, aging anxiety, and active aging.

The present study confirms the association between physical function, aging anxiety, active aging, and self-efficacy, which is consistent with previous studies [14,18,20]. Specifically, there is a significant positive correlation between physical function and active aging among older adults, which means that the higher the physical function, the higher the level of active aging. Additionally, self-efficacy has a positive correlation with physical function and active aging, indicating that high self-efficacy is related to high physical function and active aging among older adults. These results are consistent with previous studies showing that when self-efficacy is high, physical function and active aging are higher [20,31].

Furthermore, there is a significant negative correlation between aging anxiety and active aging in older adults, meaning that the higher the aging anxiety, the lower the level of active aging. Furthermore, self-efficacy has a negative correlation with aging anxiety, indicating that low self-efficacy is related to high aging anxiety in older adults. These results support previous research findings indicating that the higher the level of active aging, and the higher the self-efficacy, the lower the aging anxiety [5,21]. These results also confirm the psychological effects of active aging. Specifically, older adults can experience a high level of aging-related anxiety due to physical aging. For the active aging of older adults, considering the adverse effects of aging-related anxiety, interventions and programs to reduce aging anxiety can promote the active aging of older adults.

Finally, self-efficacy moderates the effect of physical function on active aging among older adults. This indicates that high levels of self-efficacy attenuate the negative effect of physical function deterioration on active aging among older adults. Low physical function can lead to low levels of active aging. However, if psychological support is provided to increase self-efficacy, the level of active aging can be increased. On the other hand, older adults with low self-efficacy show low levels of active aging because of decreased physical function. Furthermore, self-efficacy moderates the effect of aging anxiety on active aging. This suggests that a high level of anxiety related to aging can lead to low levels of active aging, but providing psychological support that improves self-efficacy can increase the level of active aging. Consistent with previous studies, this study confirmed that self-efficacy is a coping mechanism that moderates the effects of active aging in older adults [32].

As a result, self-efficacy supports older adults’ difficulties in activities and their ability to cope with various situations. Therefore, higher self-efficacy indicates stronger positive effects of physical function on active aging and less negative effects of aging anxiety on active aging. Older adults with a high level of physical function can face various experiences and overcome difficulties by using self-efficacy when engaging in activities [18]. In addition, older adults who experience a high level of aging anxiety may show limitations in participating in social activities or activities that promote health and safety [33]. However, even in the case of high anxiety levels, active aging can be improved if a psychological program is provided to improve self-efficacy, which is a belief in oneself [14]. In the aging process of older adults, participation in activities is related to good physical functioning and psychological stability. The weakening of physical function and aging anxiety experienced during the aging process are factors that threaten the psychological stability of older adults [34]. However, although weakening physical function and psychological anxiety in older adults affect active aging, the higher the level of self-efficacy in older adults, the better the degree of active aging. The vulnerability related to aging can cause psychological problems such as anxiety and somatization disorders [10], which can lower the self-efficacy of older adults. For this reason, the results of this study revealed the need for self-efficacy as an important internal resource for older adults to adapt to changes in old age and continue to grow in the latter part of life.

Older adults complain of difficulties in psychological well-being due to physical aging, job retirement, and the death of friends or spouses [35]. A study that examined the needs of older adults to participate in activities reported that physical activity support and psychological program support services are needed [36,37]. To increase participation in activities and improve active aging despite the functional decline and psychological anxiety of older adults, self-efficacy must be improved through appropriate service support [38]. Older people in aging societies may have lowered self-efficacy due to negative social awareness and fear of loss of meaning in life, and social support is necessary for stable old age. Specifically, integrated support for social and public services is needed. Furthermore, because they experience physical, mental, and social problems, they experience restrictions in participation in social activities; thus, they emphasize identifying their role through active activities and opportunities for new challenges in active aging [22]. For example, providing light economic activities for older adults in the community to provide them with opportunities to challenge themselves as members of society can help improve their self-efficacy.

Furthermore, in addition to physical activity programs, systematic education and training should be provided for families and caregivers through the development of programs to support psychological stability and briefing sessions to prevent degenerative diseases, the production of educational support manuals, and the revitalization of self-help groups. It is necessary to change perception through various channels to reduce the negative views of older adults and anxiety about themselves. In the aging process, older adults experience negative psychology due to changes in appearance, loneliness, lethargy, and declines in social activities. If the negative emotions of older adults are not resolved, self-anxiety deepens and leads to depression, which lowers physical function and can lead to a serious social burden in old age [38].

Therefore, as an approach for the active aging of older adults, this study suggests the need for psychological support through self-efficacy to expand opportunities for participation in activities through activation of physical functions and to overcome the aging anxiety of older adults.

This study emphasizes the necessity of a self-efficacy control on functional weakening as a method of active aging of older adults, but there are some limitations. First, the participants in this study are limited to older adults living in Gwangju and Jeollanamdo, South Korea. Therefore, care should be taken when generalizing the results of this study. Second, programs that can support older adults include various types of approaches, such as physical, cognitive, instrumental, and environmental support. However, in this study, only the support from a personal psychological control program for older adults was considered. Therefore, it may be difficult to judge whether a psychological self-efficacy support program is more effective in maintaining active aging even with the deterioration of physical function among older adults and the influence of aging anxiety. Therefore, future studies should clarify the relationship between factors related to active aging according to various type of approaches. Furthermore, research on the development of effective psychological support programs should continue.

## 5. Conclusions

This study investigated the relationships among active aging, physical function, aging anxiety, and self-efficacy among older adults. According to the results, older adults with high physical function and low aging anxiety levels experienced high levels of active aging, and support for self-efficacy could contribute to the promotion of active aging. Therefore, there is a need for the development of psychological support programs for aging adults and those with disabilities in the future.

These results suggest that a psychological support program to promote self-efficacy is an important resource as a community support system to prevent decline in occupational participation due to physical function decline and aging anxiety and to improve active aging in older people.

## Figures and Tables

**Figure 1 healthcare-13-00108-f001:**
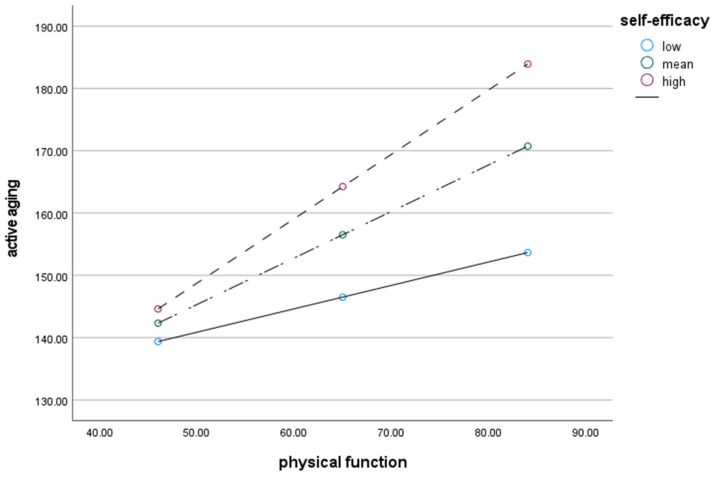
Moderating effect of self-efficacy on the relationship between physical function and active aging.

**Figure 2 healthcare-13-00108-f002:**
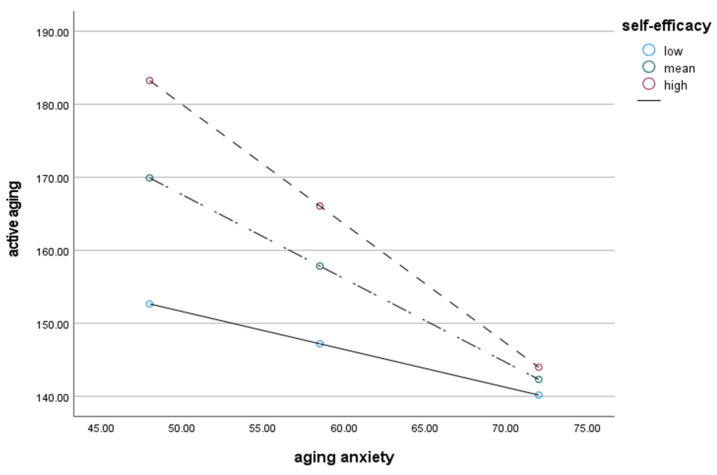
Moderating effect of self-efficacy on the relationship between aging anxiety and active aging.

**Table 1 healthcare-13-00108-t001:** Characteristics of the subjects (*N* = 280).

Classification	Number (*N*)	Percentage (%)
Sex		
Male	151	53.9
Female	129	46.1
Age		
65–69	75	26.8
70–74	71	25.4
75–79	57	20.3
80≤	77	27.5
Academic education background		
No education	12	4.3
Elementary school graduate	55	19.6
Middle school graduate	61	21.8
High school graduate	92	32.9
University graduation, and above	60	21.4
Housemate		
Solitary residence	41	14.6
Gametes	206	73.6
Children	23	8.2
Relative	10	3.6
Religion presence		
Yes	159	56.8
No	121	43.2
Economic level		
Difficulty	23	8.2
Ordinary	190	67.9
Leisureliness	67	23.9

**Table 2 healthcare-13-00108-t002:** Correlations and descriptive statistics for the measured variables (*N* = 280).

Variables	1	2	3	4	*M*	*SD*	Range
1. Physical function	1				66.20	16.86	30–105
2. Aging anxiety	−0.574 ***	1			58.82	10.02	22–80
3. Self-efficacy	0.172 ***	−0.160 **	1		59.22	10.81	33–85
4. Active aging	0.611 ***	−0.595 **	0.360 **	1	157.68	24.70	80–234

*M* = mean; *SD* = standard deviation.; 1 = Physical function; 2 = Aging anxiety; 3 = Self-efficacy; 4 = Active aging. *** *p* < 0.001, ** *p* < 0.01.

**Table 3 healthcare-13-00108-t003:** Moderating effect of self-efficacy on the relationship between physical function and active aging.

Predictor	Coefficient (*B*)	*SE*	*t*	*LLCI*	*ULCI*
Intercept (Constant)	156.789	1.081	145.027 ***	154.661	158.917
Physical function	0.725	0.068	10.471 ***	0.591	0.858
Self-efficacy	0.804	0.109	7.402 ***	0.590	1.018
Interaction	0.029	0.006	4.828 ***	0.017	0.040
F		86.492 ***			
*R* ^2^		0.485			
Δ*R*^2^		0.043 *** (F = 23.309)			

*SE* = standard error; *LLCI* = lower limit confidence interval; *ULCI* = upper limit confidence interval. Note. Interaction = Interaction between physical function and self-efficacy. *** *p* < 0.001.

**Table 4 healthcare-13-00108-t004:** Conditional effects of self-efficacy on physical function and active aging.

Self-Efficacy	Effect	*SE*	*t*	95% *CI*	
				*LL*	*UL*
Low (−1SD)	0.376	0.114	3.302 **	0.152	0.600
Mean	0.747	0.067	11.238 ***	0.616	0.878
High (+1SD)	1.034	0.077	13.434 ***	0.882	1.185

*SE* = standard error; *LL* = lower limit; *UL* = upper limit. *** *p* < 0.001, ** *p* < 0.01.

**Table 5 healthcare-13-00108-t005:** Moderating effect of self-efficacy on the relationship between aging anxiety and active aging.

Predictor	Coefficient (*B*)	*SE*	*t*	*LLCI*	*ULCI*
Intercept (Constant)	156.844	1.084	144.703 ***	154.710	158.978
Aging anxiety	−1.112	0.118	−9.392 ***	−1.345	−0.879
Self-efficacy	0.803	0.107	7.547 ***	0.594	1.013
Interaction	−0.048	0.009	−5.217 ***	−0.067	−0.030
F			84.249 ***		
*R* ^2^			0.478		
Δ*R*^2^			0.052 *** (F = 27.218)		

*SE* = standard error; *LLCI* = lower limit confidence interval; *ULCI* = upper limit confidence interval. Note. Interaction = Interaction between aging anxiety and self-efficacy. *** *p* < 0.001.

**Table 6 healthcare-13-00108-t006:** Conditional effects of self-efficacy on aging anxiety and active aging.

Self-Efficacy	Effect	*SE*	*t*	95% *CI*	
				*LL*	*UL*
Low (−1SD)	−0.520	0.194	−2.680 **	−0.902	−0.138
Mean	−1.150	0.116	−9.937 ***	−1.378	−0.922
High (+1SD)	−1.636	0.121	−13.542 ***	−1.874	−1.398

*SE* = standard error; *LL* = lower limit; *UL* = upper limit. *** *p* < 0.001, ** *p* < 0.01.

## Data Availability

The raw data supporting the conclusions of this article will be made available by the author upon reasonable request after signing a confidentiality agreement.

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
