# Peer review of "The Moderating Effect of Self-Efficacy on Physical Function, Aging Anxiety, and Active Aging in Community-Dwelling Older Adults"

_healthcare, 2025, doi:10.3390/healthcare13020108_

Round 1
Reviewer 1 Report
Comments and Suggestions for Authors
Thank you very much for giving me the opportunity to review this fabulous work. I am adding some comments to facilitate understanding and value of it:
- 27 questionnaires have been excluded for being answered insincerely. On what basis was it determined whether they had been answered insincerely?
- In the OPTIMAL Instrument, why have the scores been reversed? If the validation of the psychometric properties was done in a certain way, on what basis was that decision made? What is the alpha coefficient?
- AAS: Why are the negative items reversed?
- More should be explained about the Process macromodel, the B coefficient
A section on “outcome variables” should be added and a detailed explanation should be given of how each of them has been used, as well as the justifications that have been included in several of them when using them. If it is really considered pertinent to make this modification, the reason and how it is done should be explained in detail. The explanation of the methodology is too brief. The assessment instruments are not sufficiently explained to understand the results
- The only inclusion criterion for the sample was being 65 years old or older. Was any type of screening done for cognitive impairment, disease with repercussions on physical function, any other diagnosis?
- Table 2, in the range column, what data does it show?
- In the tables, the abbreviations should be described at the bottom of the table. SE, LLCI, ULCI are not understood
- Displaying a table with the results obtained for all the measurement instruments could help to understand the results.
- In the bibliography section, the references used about aging and physical activity are too old; they should be updated.
Author Response
Dear Reviewer 1,
Thank you very much for taking the time to review this manuscript.
Please see the attachment.
Response to Reviewer’ Comments
- Summary
Thank you very much for taking the time to review this manuscript. Please find the detailed response below and the corresponding revisions/corrections highlighted/in track changes in the re-submitted files.
- Point-by-point response to Comments and Suggestions for Authors
Comment 1: 27 questionnaires have been excluded for being answered insincerely. On what basis was it determined whether they had been answered insincerely?
Response 1: Thank you for pointing this out. We agree with this comment. We are in a situation where positive and negative questions are mixed in the characteristics of the evaluation tool used in the paper. As a result, we excluded the questionnaires that left 'non-differentiation' and 'do not know' as blank answers in a series of scale questions.
We have revised and supplemented this by adding references. – page 3, paragraph 2, lines 15-17.
“Excluding 27 questionnaires that insincerely responded (non-differentiation) or left blank, 280 were used in the final analysis [24]”
Comment 2: In the OPTIMAL Instrument, why have the scores been reversed? If the validation of the psychometric properties was done in a certain way, on what basis was that decision made? What is the alpha coefficient?
Response 2: Thank you for pointing this out. We agree with this comment. We have revised and supplemented the content of the text. – page 3, paragraph 3 & page 4, paragraph 1.
We used 'OPTIMAL instrument' as a survey tool to measure the physical function level of the older adults.
The score of the tool is a self-report measurement tool from 1 point (possible without difficulty) to 5 points (not possible) for 21 items. In order to help understand the level of physical function, the result score of the tool was reversed to the score of each item, and the researcher arbitrarily assigned the high score to indicate the good physical function level and to interpret the result.
Alpha coefficient stands for Cronbach's alpha.
Comment 3: AAS: Why are the negative items reversed?
Response 3: Thank you for pointing this out. We agree with this comment. We used the Korean version of the Aging Anxiety Scale, used to determine the level of aging anxiety in older people, is a 20-item self-report scale consisting of 13 positive items and 7 negative items. To measure a single concept of aging anxiety, negative items were inversely transformed and the overall score was reflected in statistical analysis. A lower score indicates a lower level of aging anxiety.
Comment 4: More should be explained about the Process macromodel, the B coefficient.
A section on “outcome variables” should be added and a detailed explanation should be given of how each of them has been used, as well as the justifications that have been included in several of them when using them. If it is really considered pertinent to make this modification, the reason and how it is done should be explained in detail. The explanation of the methodology is too brief. The assessment instruments are not sufficiently explained to understand the results
Response 4: Thank you for pointing this out. We agree with this comment. Therefore, we have revised and emphasize the contents of the text and the contents of 'outcome variables' in Tables 3 and 5. - page 5, paragraph 2, lines 7-11 & page 6 Table 3, paragraph 2, lines 5-11 & page 7 Table 5, paragraph 1, lines 7-11 & page 8, paragraph 1, lines 5-11.
The B coefficient, p < 0.001, LLCI and ULCI (95% confidence interval) presented in the table do not include '0', showing significant results. In the relationship between independent and dependent variables, the moderating variable is showing a moderating effect.
Comment 5: The only inclusion criterion for the sample was being 65 years old or older. Was any type of screening done for cognitive impairment, disease with repercussions on physical function, any other diagnosis?
Response 5: Thank you for pointing this out. We agree with this comment. Therefore, we have revised and emphasize the contents of the text. – page 3, paragraph 2, lines 8-10.
The purpose of this study is to propose a plan to increase the level of active aging of older adults in the community.
Therefore, older adults with cognitive impairment or difficulty in physical function due to illness or other diagnosis were excluded from the selection criteria.
“The survey was conducted on subjects who understood the research plan and wrote a written consent form, older adults with cognitive impairment or difficulty in physical function were excluded from the criteria.”
Comment 6: Table 2, in the range column, what data does it show?
Response 6: Thank you for pointing this out. We agree with this comment. Therefore, we have revised and emphasize. – page 5, Table 2.
The results in Table 2 show the correlations between the measured variables. The data displayed in the column is indicated in variables of the first column. It represents data on ‘1 = Physical function’, ‘2 = Aging anxiety’, ‘3 = Self-efficacy’, and ‘4 = Active aging’.
Comment 7: In the tables, the abbreviations should be described at the bottom of the table. SE, LLCI, ULCI are not understood
Response 7: Thank you for pointing this out. We agree with this comment. Therefore, we have revised and emphasize. – page 6, Table 3 & 4 and page 7, Table 5 and page 8 Table 6.
“SE = standard error, LLCI = lower limit confidence interval, ULCI = upper limit confidence interval, LL = lower limit, UL = upper limit”
Comment 8: Displaying a table with the results obtained for all the measurement instruments could help to understand the results.
Response 8: Thank you for pointing this out. We agree with this comment. Therefore, we have revised and emphasize. – page 5, Table 2.
Table 2, presents the correlation along with the mean, standard deviation, and measurement range of all measuring instruments.
Comment 9: In the bibliography section, the references used about aging and physical activity are too old; they should be updated.
Response 9: Thank you for pointing this out. We agree with this comment. Therefore, we have revised and supplemented it by adding up-to-date references on aging and physical function.
– page 2, paragraph 1, lines 6-8 & paragraph 2, lines 3-5.
“an active lifestyle can mitigate the aging process [8], and physically active older adults have better health and vitality [9].”
“Aging anxiety is a negative emotion associated with aging, including the fear of aging and feelings of physical, mental, and personal loss [11].”
Thank you for your careful review.

Reviewer 2 Report
Comments and Suggestions for Authors
Dear Authors
It has been a pleasure to have the opportunity to review your manuscript “The Moderating Effect of Self-Efficacy on Physical Function, Aging Anxiety, and Active Aging in Community-dwelling Older Adults”
The manuscript requires major revisions.
Since the document does not have line numbers, I will try to indicate as accurately as possible the text that needs modification.
Here are some recommendations that may help to improve the manuscript:
ABSTRACT
1. “Older adults who are aging experience decreased physical function, and psychological anxiety” This statement can be interpreted to mean that anxiety always decreases with aging, but that is not the case. In fact, the introduction section states that 'In relation to aging, depression, anxiety, and dementia are likely to cause psychological problems, which can deteriorate the quality of life of older adults [10],' implying that the presence or increase of depression, dementia, and anxiety deteriorates the quality of life.
2. Method: It does not include the type of study or the type of sampling used. What was the study population? Was a sample size calculation performed? Inclusion and exclusion criteria? Please indicate the authorization number of the research committee.
3. Results: Include the participation sample in this section and remove it from the previous section.
INTRODUCTION
4. “Aging anxiety is correlated with self-efficacy, self-fulfillment, and successful aging”. Review this statement, I believe it is not accurate. Could it be that you meant to indicate a negative correlation?"
MATERIAL Y METHODS
5. This section begins with “2.1 study participants”. However, it is necessary to include before this: type of study, type of sampling, study population, time period, and authorization number.
Subsequently, in the study participants section, include the sample (based on the sample size equation) that is expected to be captured to be representative. Inclusion and exclusion criteria.
You can add another subsection participant recruitment procedure Here, detail the entire procedure from dissemination of the study through brochures, the freedom and autonomy of the person who contacts to participate, the information provided by the researchers, the 'verbal' acceptance required for the subsequent sending of documentation (consent forms, questionnaires, etc.).
RESULTS
6. “3.1. Demographic data. Our study included 280 older people living in Gwangju and Jeollanamdo, South Korea” Is it a representative sample of the population?
7. “Furthermore, aging anxiety was negatively correlated with self-efficacy and active aging (r = -.160, r = .595, p < 0.01), whereas self-efficacy was positively correlated with active aging (r = .360, p < 0.01)” Please review the data (.595 à -.595).
8. "Table 2”: review column 2, the data -160** ( -.160**) and -595** (-.595**)
9. "3.5. Moderating Effects of Self-Efficacy on Aging Anxiety and Active Aging". "The moderating effect of self-efficacy was significant in the relationship between aging anxiety and active aging (B =-0.048, p < 0.001)". please correct 0.048 à .048
DISCUSION
10. Pagina 9. “First, the participants in this study are limited to older adults living in Gwangju and Jeollanamdo, South Korea. Therefore, care should be taken when generalizing the results of this study”
As mentioned in the methodology section, it is necessary to know the study population and the minimum required sample size (sample size equation) that allows us to assert with 280 participants that these results are limited to the population of older adults living in Gwangju and Jeollanamdo, South Korea, as this may not be the case.
Good luck with your article.
Author Response
Dear Reviewer 2,
Thank you very much for taking the time to review this manuscript.
Please see the attachment.
Response to Reviewer’ Comments
- Summary
Thank you very much for taking the time to review this manuscript. Please find the detailed response below and the corresponding revisions/corrections highlighted/in track changes in the re-submitted files.
- Point-by-point response to Comments and Suggestions for Authors
Comment 1: Abstract: “Older adults who are aging experience decreased physical function, and psychological anxiety” This statement can be interpreted to mean that anxiety always decreases with aging, but that is not the case. In fact, the introduction section states that 'In relation to aging, depression, anxiety, and dementia are likely to cause psychological problems, which can deteriorate the quality of life of older adults [10],' implying that the presence or increase of depression, dementia, and anxiety deteriorates the quality of life.
Response 1: Thank you for pointing this out. We agree with this comment. We checked the meaning of the sentences and revised and emphasize the content of the text. – page 1, paragraph 1, lines 1-2.
“Aging older adults experience psychological anxiety along with declines in physical function, which decreases the level of active aging and physical activity participation.”
Comment 2: Method: It does not include the type of study or the type of sampling used. What was the study population? Was a sample size calculation performed? Inclusion and exclusion criteria? Please indicate the authorization number of the research committee.
Response 2: Thank you for pointing this out. We agree with this comment. Therefore, we have revised and emphasize the contents of the text. – page 1, paragraph 1, lines 4-5.
“The subjects were 280 older adults living in the community.”
Comment 3: Results: Include the participation sample in this section and remove it from the previous section.
Response 3: Thank you for pointing this out. We agree with this comment. Therefore, we have revised and emphasize the contents of the text. – page 1, paragraph 1, lines 9-10.
“A total of 280 participants (aged 65–92 years) in Gwangju and Jeollanamdo, South Korea, completed a set of self-reported measures.”
Comment 4: “Aging anxiety is correlated with self-efficacy, self-fulfillment, and successful aging”. Review this statement, I believe it is not accurate. Could it be that you meant to indicate a negative correlation?"
Response 4: Thank you for pointing this out. We agree with this comment. Therefore, we checked the meaning of the sentences, and modified and emphasize the contents of the text.
– page 2, paragraph 4, lines 4-5.
“Aging anxiety negatively affects self-efficacy, self-fulfillment, and successful aging.”
Comment 5: This section begins with “2.1 study participants”. However, it is necessary to include before this: type of study, type of sampling, study population, time period, and authorization number.
Subsequently, in the study participants section, include the sample (based on the sample size equation) that is expected to be captured to be representative. Inclusion and exclusion criteria.
Response 5: Thank you for pointing this out. We agree with this comment. Therefore, we have revised and supplemented the contents of the text. – page 3, paragraph 1, lines 1-6 and paragraph 2, lines 7-9.
“2.1. Type of study
This study was a cross-sectional study design to examine the relationships between physical function, aging anxiety, active aging, and self-efficacy. This design allowed analyze the influence and moderating effects between variables. The data was collected through questionnaires, from August 2022 to March 2023. This study was approved by the Institutional Review Board of Dongshin University (IRB No. 201905-SB-017).”
“2.2. Study Participants and recruitment procedure
The survey was conducted on subjects who understood the research plan and wrote a written consent form, older adults with cognitive impairment or difficulty in physical function were excluded from the criteria”
Comment 6: “3.1. Demographic data. Our study included 280 older people living in Gwangju and Jeollanamdo, South Korea” Is it a representative sample of the population?
Response 6: Thank you for pointing this out. We agree with this comment. The results of this study consisted of a sample group from a limited region, which suggested limitations in “Discussion”. To complement this point, later research will focus on a nationwide sample group.
Comment 7: “Furthermore, aging anxiety was negatively correlated with self-efficacy and active aging (r = -.160, r = .595, p < 0.01), whereas self-efficacy was positively correlated with active aging (r = .360, p < 0.01)” Please review the data (.595 à -.595).
Response 7: Thank you for pointing this out. We agree with this comment. Therefore, we have revised and emphasize. – page 5, paragraph 1, line 8.
Comment 8: "Table 2”: review column 2, the data -160** ( -.160**) and -595** (-.595**)
Response 8: Thank you for pointing this out. We agree with this comment. Therefore, we have revised and emphasize. – page 5, Table 2.
Comment 9: "3.5. Moderating Effects of Self-Efficacy on Aging Anxiety and Active Aging". "The moderating effect of self-efficacy was significant in the relationship between aging anxiety and active aging (B =-0.048, p < 0.001)". please correct 0.048 à .048.
Response 9: Thank you for pointing this out. We agree with this comment. Therefore, we have revised and emphasize. – page 7, paragraph 1, line 7.
Comment 10: Pagina 9. “First, the participants in this study are limited to older adults living in Gwangju and Jeollanamdo, South Korea. Therefore, care should be taken when generalizing the results of this study”
Response 10: Thank you for pointing this out. We agree with this comment. The results of this study consisted of a sample group from a limited region, which suggested limitations in “Discussion”. In future studies, we will focus on a nationwide sample group. – page 3, paragraph 2, lines 11-16.
“G*Power analysis was used to calculate the sample size for the study [23]. Two independent variables, one dependent variable, and one moderator variable were set, and a total sample size of 129 was calculated, and the required number of subjects was applied.”
Thank you for your careful review.

Reviewer 3 Report
Comments and Suggestions for Authors
This study examined the relationship between physical function, ageing anxiety, self-efficacy and active ageing in older people aged 65 and over. Although the results are largely in line with what might be expected and do not have a major impact, they may have some value in showing a significant relationship between the concept of active ageing, which has been attracting attention in recent years, and conventional other positive and negative indicators.
Although the results of the survey were based on a self-administered questionnaire, so the existence of bias cannot be fully denied, there are no particular statistical problems, so the paper can be considered to be publishable with some value.
One minor correction to be required is that the ranges (minimum and maximum values) in Table 2 and in the text were written to two decimal places in the form of .00, but this has no meaning as a significant figure and should be omitted.
Author Response
Dear Reviewer 3,
Thank you very much for taking the time to review this manuscript.
Please see the attachment.
Response to Reviewer’ Comments
- Summary
Thank you very much for taking the time to review this manuscript. Please find the detailed response below and the corresponding revisions/corrections highlighted/in track changes in the re-submitted files.
- Point-by-point response to Comments and Suggestions for Authors
Comment 1: Although the results of the survey were based on a self-administered questionnaire, so the existence of bias cannot be fully denied, there are no particular statistical problems, so the paper can be considered to be publishable with some value.
Response 1: Thank you for pointing this out. We agree with this comment. Therefore, we have revised and emphasize the contents of the text. – page 3, paragraph 2, lines 14-16.
“Excluding 27 questionnaires that insincerely responded (non-differentiation) or left blank, 280 were used in the final analysis [24].”
Comment 2: One minor correction to be required is that the ranges (minimum and maximum values) in Table 2 and in the text were written to two decimal places in the form of .00, but this has no meaning as a significant figure and should be omitted.
Response 2: Thank you for pointing this out. We agree with this comment. Therefore, we have revised and emphasize the contents of the text. – page 5, Table 2.
Thank you for your careful review.

Round 2
Reviewer 2 Report
Comments and Suggestions for Authors
Dear Authors
After reviewing the changes made by the authors in the manuscript, there is one aspect that needs to be reviewed again, especially because it is a methodological aspect:
In the previous review, I recommended: “This section begins with “2.1 study participants”. However, it is necessary to include before this: type of study, type of sampling, study population, time period, and authorization number. Subsequently, in the study participants section, include the sample (based on the sample size equation) that is expected to be captured to be representative. Inclusion and exclusion criteria. You can add another subsection participant recruitment procedure Here, detail the entire procedure from dissemination of the study through brochures, the freedom and autonomy of the person who contacts to participate, the information provided by the researchers, the 'verbal' acceptance required for the subsequent sending of documentation (consent forms, questionnaires, etc.).
· They do not indicate the type of sampling applied.
· On the contrary, a sample calculation of 129 is included (they do not indicate the level of confidence, reference population, estimate of the proportion in the population, replacement rate if there is one… ….).
· They state that 27 questionnaires were excluded, with the final sample being 280 [lines 114-118]. It is necessary to clarify that the total number of participants amounted to 307 subjects who were recruited through published flyers (methodology section), however, in the results section, the final sample after exclusion, it should be indicated that the sample amounted to 280 subjects.
· When the mentioned changes are made, make the pertinent corrections in the abstract (methodology). Include the type of study and the total number of participants in the methods.
Although only a few corrections are requested, they are relevant to the manuscript, so I have indicated that it requires major revisions.
Greetings
Author Response
Dear Reviewer 2,
Thank you very much for your detailed comments and taking the time to re-review this manuscript. Please find the detailed response below and the corresponding revisions/corrections highlighted/in track changes in the re-submitted files.
Point-by-point response to Comments and Suggestions for Authors
Comment 1: This section begins with “2.1 study participants”. However, it is necessary to include before this: type of study, type of sampling, study population, time period, and authorization number. Subsequently, in the study participants section, include the sample (based on the sample size equation) that is expected to be captured to be representative. Inclusion and exclusion criteria. You can add another subsection participant recruitment procedure Here, detail the entire procedure from dissemination of the study through brochures, the freedom and autonomy of the person who contacts to participate, the information provided by the researchers, the 'verbal' acceptance required for the subsequent sending of documentation (consent forms, questionnaires, etc.).
- They do not indicate the type sampling applied.
- It is necessary to clarify that the total number of participants amounted to 307 subjects who were recruited through published flyers (methodology section), however, in the results section, the final sample after exclusion, it should be indicated that the sample amounted to 280 subjects.
Response 1: Thank you for pointing this out. We agree with this comment. Therefore, we have revised and supplemented the contents of the text. – page 3, paragraph 1, paragraph 2, and paragraph 3.
“2.1. Type of study
This study was a cross-sectional study design to examine the relationships between physical function, aging anxiety, active aging, and self-efficacy. The type of sampling used to collect data was the snowball sampling. The subjects were older adults aged 65 or older living in Gwangju and Jeollanamdo, and the data was collected through questionnaires, from August 2022 to March 2023. This study was approved by the Institutional Review Board of Dongshin University (IRB No. 201905-SB-017).”
“2.2. Study Participants
Participants were recruited via flyers posted. The number of sample groups was calculated using the G* power 3.1 program, with effect size of .15, significance probability of .05, power of 95%, and 3 independent variables, the minimum required sample size was determined 129. The total number of participants collected through the published flyers was 307, however, 27 questionnaires including insincere responses and missing values were excluded, and 280 were used in the final analysis [24]. The survey was conducted on subjects who understood the research plan and wrote a written consent form, older adults with cognitive impairment or difficulty in physical function were excluded from the criteria.”
“ 2.3. Participant recruitment procedure
Participants in this study were recruited primarily by sending out informational materials containing the purpose, subjects, content, procedures, and consent forms to senior health centers, senior welfare centers, and public health centers in Gwangju and Jeollanamdo, South Korea. The researcher visited the institution to obtain written consent from the participants, explained the study, and conducted a survey with the participants who agreed to participate in the study based on their personal freedom and autonomy, obtaining their signatures.”
Comment 2: When the mentioned changes are made, make the pertinent corrections in the abstract (methodology). Include the type of study and the total number of participants in the methods.
Response 2: Thank you for pointing this out. We agree with this comment. Therefore, we have revised and emphasize. - page 1, paragraph 1, lines 4-8 and lines 10-11.
“Methods: A cross-sectional survey was conducted among older adults in Gwangju and Jeollanamdo using snowball sampling method. The final sample consisted of 280. The survey included demographic data, Outpatient Physical Therapy Improvement in Movement Assessment Log, Anxiety about Aging Scale, General Self-Efficacy Scale, and Active Aging Scale, respectively.”
“Results: A total of 307 participants (aged 65–92 years), and 280 data were used in the final analysis after excluding 27 incomplete data.”
Thank you for your careful review.
